# Transfer of POCUS Skills of Anesthesia Trainees from the Simulation Laboratory to Clinical Practice: A Follow-Up Pilot Evaluation After ABC US Protocol Training

**DOI:** 10.3390/diagnostics15030354

**Published:** 2025-02-03

**Authors:** Robert Simon, Cristina Petrișor, Constantin Bodolea, Oana Antal, Marta Băncișor, Orlanda Moldovan, Ion Cosmin Puia

**Affiliations:** 1Doctoral School, Faculty of Medicine, University of Oradea, 410087 Oradea, Romania; 2Anesthesia and Intensive Care Department, “Iuliu Hatieganu” University of Medicine and Pharmacy, 400347 Cluj-Napoca, Romania; 3Clinical Institute of Urology and Renal Transplantation, 400000 Cluj-Napoca, Romania; 4Clinical County Emergency Hospital, 400347 Cluj-Napoca, Romania; 5Municipal Clinical Hospital, 400139 Cluj-Napoca, Romania; 6Clinical Emergency Hospital for Children, 400177 Cluj-Napoca, Romania; 7Regional Institute of Gastroenterology and Hepatology, 400394 Cluj-Napoca, Romania; 8Surgery Department, “Iuliu Hatieganu” University of Medicine and Pharmacy, 400347 Cluj-Napoca, Romania

**Keywords:** point-of-care ultrasound, airway ultrasound, lung ultrasound, cardiac ultrasound, simulation-based medical education and training, point-of-care ultrasound in the intensive care unit, transition from simulation to practice, point-of-care ultrasound teaching, point-of-care ultrasound simulation

## Abstract

**Background/objectives:** Point-of-care ultrasound (POCUS) in the intensive care unit (ICU) has gained much attention in the last few years as an alternative to the classic ways of assessing and diagnosing life-threatening conditions in critical patients. During the COVID-19 pandemic, we proposed a POCUS protocol based on the airway, breathing, and circulation (ABC) approach to quickly evaluate and diagnose life-threatening diseases in critical patients with acute respiratory failure and shock, and later, we used it as a curriculum to teach POCUS to anesthesia and intensive care trainees. **Methods:** We developed an evaluation protocol where evaluators with experience in POCUS in critically ill patients had to assess the trainee’s ultrasound scan; this was based on the ABC protocol taught in the simulation laboratory and applied in a clinical setting at the bedside. **Results:** Statistically significant differences were observed in some categories evaluated regarding independence and diagnosis. **Conclusion:** Initial POCUS simulation-based training using an ABC POCUS protocol (that demonstrated good results in the simulation laboratory) is useful when transferring US skills to the bedside and is applicable in daily clinical practice with good results in terms of operator independence.

## 1. Introduction

Point-of-care ultrasound (POCUS) in the intensive care unit (ICU) has gained much attention in the last years as an alternative to the classic ways of assessing and diagnosing life-threatening conditions in critical patients, especially during the COVID-19 pandemic, when the focus on lung ultrasound and cardiac ultrasound, performed in the ICU, was believed to improve patient outcomes [1,2,3,4].

POCUS education during training can increase its use in clinical practice and simulation-based education and training (SBET), which can aid in significant improvement in knowledge and practical skills [5].

During the COVID-19 pandemic, we proposed a POCUS protocol based on the airway, breathing, and circulation (ABC) approach to quickly evaluate and diagnose life-threatening diseases in critical patients with acute respiratory failure and shock that can substitute classical diagnosis tools [6]. Based on this protocol, we developed a curriculum for SBET in POCUS, which is useful in teaching trainees the basics of airway, lung, and cardiac POCUS in critical patients and evaluating life-threatening conditions [7].

Implementing a POCUS protocol in the ICU could lead to fewer irradiating procedures performed, with fewer risks associated with transportation, less time on mechanical ventilation, shorter hospital stays, and lower hospitalization costs [8].

The transition from an SBET setting to clinical practice can be challenging, and competence obtained on a high-fidelity simulator may not be applicable in clinical practice [9]. The transfer of US skills from the simulation laboratory to clinical practice has not yet been extensively assessed.

The aim of this study is to evaluate the clinical performance of ultrasound in the ICU by anesthesia and intensive care trainees and whether previous simulation training leads to the satisfactory use of POCUS at the bedside. We evaluated the use of equipment, scanning techniques, diagnosis accuracy, level of independence, and the time needed to perform the scan.

## 2. Materials and Methods

### 2.1. Protocol Development

We developed an evaluation protocol where evaluators with experience in POCUS in critically ill patients had to assess the trainee’s ultrasound scan based on the ABC protocol taught in the simulation laboratory and applied in the clinical setting at the bedside. The ABC protocol is aimed at diagnosing correct endotracheal tube placement for the airway part; the breathing part focuses on assessing signs of pneumothorax, pleural effusion, interstitial syndrome, and lung consolidation; the circulation part focuses on an evaluation of the heart using transthoracic echocardiography (Philips Affiniti 70, Philips, Amsterdam, The Netherlands, and Hitach Arietta 50, Hitachi, Tokyo, Japan) to diagnose severe left ventricle dysfunction, cardiac tamponade, indirect signs of pulmonary embolism, as well as an evaluation of inferior vena cava. The trainees had to verbally confirm their findings during the scanning process. For the airway, they had to confirm the presence of the tracheal tube in the correct position for intubated patients or what would be the expected sign (double lumen with a hyperechoic shadow) in case of oesophageal intubation. For the lung evaluation, they were expected to evaluate 3 lung points on either side of the thorax, described by Lichtenstein [10], and to confirm the presence or absence of a lung slide, seashore, or bar code sign using M-mode, presence of B-lines and their number, A-lines, lung point, as well as signs of pleural effusion and lung consolidation. The cardiac evaluation consisted of an evaluation of the heart using the 4 following views: subcostal, apical 4 chambers, parasternal long axis, parasternal short axis, and also an evaluation of inferior vena cava (the anteroposterior diameter and its collapse with respiratory phases). Regarding the cardiac evaluation, they had to describe the global function of the left ventricle and assess it qualitatively as being normal or severely depressed, the presence of cardiac tamponade, signs of acute right ventricle failure (an enlarged right ventricle that is qualitatively bigger than the left ventricle, with an abnormal septum that is bulging towards the left ventricle, with an enlarged inferior vena cava without collapse), and an inferior vena cava evaluation regarding the anteroposterior diameter and collapse with respiratory phases.

For each part of the protocol, airway, breathing, and circulation, the evaluators had to assess the following and provide a point from 1 to 5 as follows:

Correct use of equipment:Point: the trainee cannot correctly use the equipment, does not apply the ultrasound gel, and does not optimize the image regarding depth, gain, and zoom.Points: the trainee can correctly use the equipment with assistance.Points: the trainee manages to use the equipment with minimum assistance.Points: the trainee manages to use the equipment with some assistance; for example, verbal indications on what needs to be optimized.Points: the trainee manages to correctly use the equipment without assistance.

Structured approach: this section was used to evaluate only the pleuro-pulmonary and cardiac evaluation part of the protocol.

Point: the trainee does not have a structured approach to the examination.Points: the trainee has a structured approach with assistance.Points: the trainee has a structured approach with minimum assistance, like verbal indications and physical intervention.Points: the trainee has a structured approach with some verbal assistance.Points: the trainee has a structured approach without any assistance.

Structure recognition:Point: the trainee does not recognize the structures examined.Points: the trainee recognizes the structures if explained and shown to them on the screen.Points: the trainee recognizes the structures examined with minimal verbal assistance and minimal on-screen indications.Points: the trainee recognizes the structures examined with minimal verbal assistance.Points: the trainee recognizes the structures examined without assistance.

Diagnostic:Point: the trainee does not manage to establish an ultrasound diagnosis and does not correlate the findings with the clinical status of the patient.Points: the trainee manages to partially establish an ultrasound diagnosis and can partially correlate the findings with the clinical status of the patient with assistance.Points: the trainee manages to establish an ultrasound diagnosis and can correlate the findings with the clinical status of the patient with minimum assistance.Points: the trainee manages to establish an ultrasound diagnosis without proper correlation with the clinical status of the patient without assistance.Points: the trainee manages to establish an ultrasound diagnosis and can correlate the findings with the clinical status of the patient without assistance.

Level of independence:Point: there is a constant need for verbal and physical assistance to perform the scan.Points: there is an inconstant need for verbal and physical assistance to perform the scan.Points: there is a need for minimum verbal and/or physical assistance to perform the scan.Points: there is a need for minimum verbal assistance to perform the scan.Points: there is no need for assistance to perform the scan.

We recorded the diagnosis made by the trainee, the preexisting diagnosis of the patient, if other imagistic procedures were used, and the time needed to apply the protocol and complete the scan in minutes. The evaluators were not aware if the trainees attended prior simulation sessions, as this information was recorded after the evaluation.

### 2.2. Ethics Committee

After establishing the evaluation protocol, it was proposed to the ethics committee from the Emergency County Hospital, Cluj-Napoca, and ethics approval was received (Approval document no. 29633/28 June 2023). Patient consent was waived because the procedure was non-invasive, supervised by experienced consultants, and was not followed by medical intervention decided by the trainee who performed the scan, and the waiver was accepted by the ethics committee.

The scans were performed in the ICU of the Anaesthesia and Intensive Care 1 department at Emergency County Hospital, Cluj-Napoca, and in the ICU compartment at the Clinical Institute of Urology and Renal Transplantation, Cluj-Napoca, between October 2023 and October 2024.

### 2.3. Statistical Analysis

After an evaluation of the scans performed, we centralized all scores for each category evaluated and divided them into two groups: scans that were performed by trainees who attended the simulation sessions and those who did not attend the simulation sessions, using the Excel database. Data were described using means and standard deviation. Normal distribution was assessed in each category of the scans performed by the trainees who attended prior simulation sessions and those who did not use the coefficient of variation, variance, and Shapiro–Wilk. Because the data were not normally distributed, a Mann–Whitney U test was performed to test if there was any difference between the scores of the scans performed by the two groups, the simulation group and the no-simulation group, for each category evaluated. The same statistical test was applied to see if there was any difference between the duration of scans performed by the trainees who attended and those who did not attend prior simulation sessions. The statistical tests were performed in Excel (Version 16.93.1) and Jasp (Version 0.19.1).

## 3. Results

Between October 2023 and October 2024, one hundred scans were evaluated by three intensive care physicians with experience in POCUS in critical patients. Out of the 100 scans, 78 were included in the final statistical analysis due to errors in data collection by the evaluators, from which 63 evaluations were performed by trainees who previously attended simulation sessions and 15 by trainees who had no prior simulation with the protocol curriculum but who were informed about the structure of the protocol, as seen in Figure 1.

Figure 2 highlights the scores for each category evaluated, with the scores being represented on the vertical axis and the number of times the score was obtained on the horizontal axis. Most of the ultrasound evaluations performed by the trainees who attended prior simulation sessions had scores higher than 3 in most of the categories examined. The descriptive statistics can be observed in Table 1.

Comparative statistics results and the Mann–Whitney U test can be observed in Table 2, where we found a statistically significant difference in the level of independence regarding airway and cardiac evaluation and a significant difference in the lung diagnosis category.

## 4. Discussion

Traditionally, ultrasound was used in anesthesia and the ICU for procedural guidance and less as a diagnosis or monitorization tool used by the intensive care physician. Lung ultrasound was promoted as an alternative to classical diagnosis tools with good results [10,11] and with growing clinical use [12].

There is international evidence and recommendations for the use of ultrasound in the ICU by intensive care physicians for diagnosing, monitoring, and procedural guidance [13,14,15,16,17,18]. Using specific protocols for ultrasound-driven diagnostics and intervention has been shown to improve patient outcomes, with reduced mechanical ventilation times, reduced number of irradiation procedures, reduced hospitalization days, and reduced costs [8]. Bedside ultrasound in the ICU may be used to improve a patient’s prognosis in patients with acute respiratory failure [19] and improve the clinical outcomes in patients with sepsis [20]. There is heterogeneity of POCUS use in the ICU in Europe and around the world. Integration of ultrasonography scanning by intensive care specialists in the ICU seems to be difficult, and the use of POCUS should be based on a competence-based approach rather than user experience [21].

The utility of a standardized approach to POCUS evaluation of a patient using a protocol in the critical setting may be helpful and often change the outcome of a patient [8]. We described earlier that in a simulation-based setting, trainees who follow a simulation curriculum based on our earlier proposed protocol [6] could accurately diagnose life-threatening conditions in the critical patient with acute respiratory failure or shock within 5 min [7].

A study that assessed the use of POCUS in the ICU found that at least one-third of the patients benefited from POCUS, with heart and lung evaluations being the first two targeted evaluation organs, with an impact on diagnosis or therapeutic procedures in 85% of cases where POCUS was performed. It was concluded that POCUS performed in an emergency setting by an ICU physician with routine POCUS use has an increased impact [22]. Other data confirm that when using a POCUS protocol with an educational curriculum, new diagnoses can be set in almost two-thirds of the patients, with a change in therapeutic management in almost one-third of the patients that were scanned. In this study, most POCUSs were used for lung evaluation, followed by cardiac evaluation [23]. It has been demonstrated that the use of POCUS improves the diagnosis of acute trauma when assessing for life-threatening conditions. Narrowing the critical diagnosis and confirmation of a suspected diagnosis can be of use in an emergency. One study demonstrated that resuscitation efforts and patient outcomes are significantly influenced by POCUS use in the emergency department, where POCUS was used for an assessment of potential reversible causes of cardiac arrest [24]. Therapeutic changes and interventions after the use of POCUS are supported by other findings [8].

The transition from a simulation setting to clinical practice can be challenging, and the utility of prior training and competency on a high-fidelity simulator can ensure competency in clinical practice remains unclear. Although statistically significant differences between evaluations performed by trainees who have attended prior simulation training with the curriculum based on an ABC approach protocol exist in a few categories evaluated, on most criteria, the minimum score obtained was higher in the simulation group than in the non-simulation group, with a higher mean. The level of independence regarding airway confirmation and cardiac evaluation was higher in the group with prior simulation. In the case of the lung evaluation, the level of independence was higher in the simulation group, without statistical significance, but the scores obtained on the diagnosis criteria were higher in the simulation group. Other studies showed similar results regarding the cardiac POCUS evaluation, where most residents were able to recognize structures, but the interpretation skills obtained lower scores at evaluation. In the same setting, educational interventions had little association with the trainees’ performances and scores obtained at evaluation [25].

The duration of each evaluation was longer in clinical practice then was in a simulation setting, where each trainee was allocated 5 min for the evaluation [7]. The maximum scan time in the evaluations performed by the trainees who attended the prior simulation setting was 4 min lower than those performed by the trainees who had no prior simulation, with a mean lower than 3 min. Similar times needed to perform a POCUS based on the protocol in the ICU were found in other studies, where the protocol was applied in 15–30 min [8]. One study found that novice ultrasound users can obtain a mastery level in a simulation-based setting and can achieve good scores for evaluations with improved evaluation times after obtaining a satisfactory level using a protocol, but large variations exist in the amount of training needed [26]. Our data show that the total duration of the evaluation was lower in the evaluations performed by trainees who had prior simulation experience. This could be due to multiple factors like having more confidence in their skills and having performed multiple scans on their own or being supervised after attending the simulation sessions. Being used to the protocol approach may have improved the scanning times.

In our current study, we found a difference between the two groups regarding the level of independence in airway and cardiac evaluations and a difference regarding lung diagnosis. Even if the differences in other criteria were not statistically significant, the level of independence may have a clinical significance, which may lead the trainees to use more POCUS in their clinical practice in critical patients. Simulation training can improve self-perceived confidence and clinical performance, with existing evidence that supports the use of simulation techniques and training in anesthesia and intensive care training programs [27].

Training in POCUS on a high-fidelity simulator can aid physicians in improving their diagnostic confidence and precision [28]. A transition from the simulation laboratory to clinical practice can be challenging. Although a high-fidelity simulator for ultrasound can be useful in achieving POCUS skills, there are many factors to be taken into consideration when performing these skills in daily practice, especially in an emergency setting. The simulation setting and environment can be arranged in such a way as to mimic daily hospital conditions, but some real-life factors cannot be replicated. Factors like time limitations, emergency settings, lack of logistics, and non-technical skills applied in real life must be taken into consideration when talking about the transition from a simulation setting to real-life use. Although we previously demonstrated the usefulness of high-fidelity simulations using the ABS protocol for fast life-threatening diagnoses using POCUS, in our current follow-up study, the group who previously had simulation experience obtained higher scores with statistical differences in some categories, with the most important one being the level of independence. How these results translate to clinical practice is unsure, but having a higher independence might suggest that POCUS may be used more often, especially in unsupervised settings. One study, with a novel approach to POCUS use in cardiac resuscitation, concluded that having a simulation intervention for anesthesia trainees can improve diagnosis accuracy and treatment options and help integrate POCUS findings into clinical scenarios [29].

It was demonstrated that improvement in knowledge and skills can be seen when using high-fidelity simulation compared to low-fidelity simulation in resuscitation or traditional training regarding resuscitation [30]. Retainment of skills, knowledge, and clinical application remains an important outcome of educational intervention for which data are scarce. Constant simulation sessions, close follow-ups, and assisted clinical practice may improve the transition from the simulation setting to the clinical use of POCUS with better retention of knowledge and skills [31].

Our findings suggest satisfying results regarding the use of equipment, technique, and interpretation of images, as well as establishing a diagnosis when POCUS was performed by trainees who attended the simulation sessions; however, integrating these findings in the overall management of the patient and therapeutic decisions was not evaluated.

The limitation of our single center, follow-up pilot study is that the number of trainees that attended the simulation sessions, as well as the availability to participate in this follow-up study, is modest, and the low number of POCUS evaluations performed by trainees that had not attended the simulation sessions is limited, as well.

## 5. Conclusions

The initial POCUS simulation-based training, using an ABC POCUS protocol [6] that demonstrated good results in the simulation laboratory [7], is useful when transferring US skills to the bedside and is applicable in daily clinical practice with good results in terms of operator independence. The transition from the simulation laboratory to clinical practice may be made gradually with possible continuous learning interventions and supervision. Future research is needed to confirm the influence on retainment of skills, knowledge, and patient outcomes.

## Figures and Tables

**Figure 1 diagnostics-15-00354-f001:**
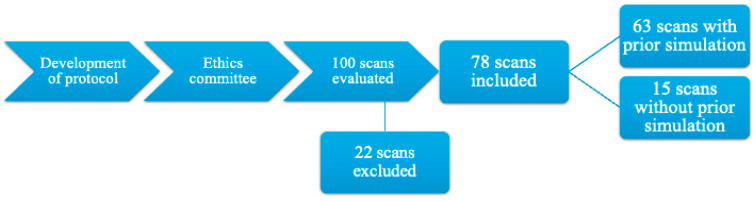
Flowchart with the study timeline, number of evaluations performed, and final number for analysis.

**Figure 2 diagnostics-15-00354-f002:**
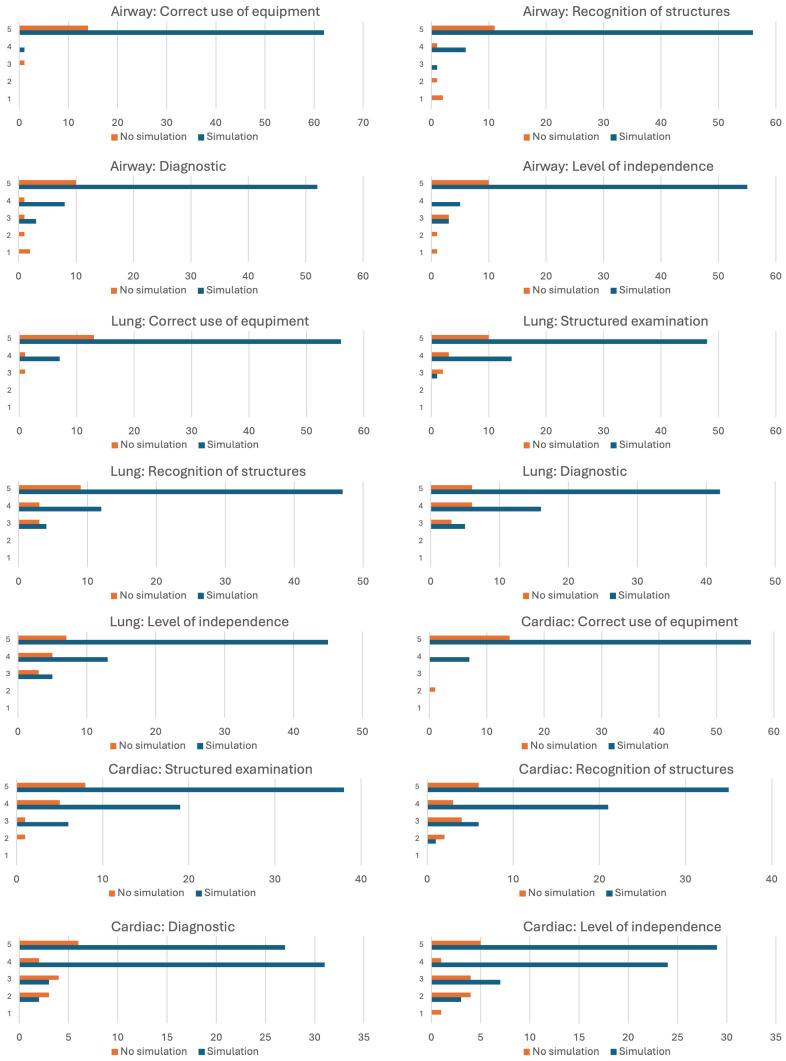
Scores attributed for each category assessed.

**Table 1 diagnostics-15-00354-t001:** Descriptive statistics.

	Simulation	Mean	Std. Dev.	Coeff. of Variation	Variance	Shapiro–Wilk	*p*-Value of Shapiro–Wilk
Airway: Correct use of equipment	yes	4.984	0.126	0.025	0.016	0.106	<0.001
	no	4.867	0.516	0.106	0.267	0.284	<0.001
Airway: Recognition of structures	yes	4.873	0.381	0.078	0.145	0.372	<0.001
	no	4.200	1.521	0.362	2.314	0.576	<0.001
Airway: Diagnostic	yes	4.778	0.522	0.109	0.272	0.476	<0.001
	no	4.067	1.534	0.377	2.352	0.654	<0.001
Airway: Level of independence	yes	4.825	0.493	0.102	0.243	0.397	<0.001
	no	4.133	1.356	0.328	1.838	0.685	<0.001
Lung: Correct use of equipment	yes	4.889	0.317	0.065	0.100	0.364	<0.001
	no	4.800	0.561	0.117	0.314	0.421	<0.001
Lung: Structured examination	yes	4.746	0.474	0.100	0.225	0.555	<0.001
	no	4.533	0.743	0.164	0.552	0.663	<0.001
Lung: Recognition of structures	yes	4.683	0.591	0.126	0.349	0.577	<0.001
	no	4.400	0.828	0.188	0.686	0.705	<0.001
Lung: Diagnostic	yes	4.587	0.638	0.139	0.408	0.653	<0.001
	no	4.200	0.775	0.184	0.600	0.806	0.004
Lung: Level of independence	yes	4.635	0.630	0.136	0.397	0.610	<0.001
	no	4.267	0.799	0.187	0.638	0.783	0.002
Cardiac: Correct use of equipment	yes	4.889	0.317	0.065	0.100	0.364	<0.001
	no	4.800	0.775	0.161	0.600	0.284	<0.001
Cardiac: Structured examination	yes	4.508	0.669	0.148	0.448	0.700	<0.001
	no	4.333	0.900	0.208	0.810	0.748	<0.001
Cardiac: Recognition of structures	yes	4.429	0.734	0.166	0.539	0.737	<0.001
	no	3.867	1.125	0.291	1.267	0.840	0.012
Cardiac: Diagnostic	yes	4.317	0.714	0.165	0.510	0.748	<0.001
	no	3.733	1.223	0.328	1.495	0.823	0.007
Cardiac: Level of independence	yes	4.254	0.842	0.198	0.709	0.783	<0.001
	no	3.333	1.397	0.419	1.952	0.863	0.027
Duration of examination	yes	13.476	4.724	0.351	22.318	0.952	0.015
	no	16.200	6.971	0.430	48.600	0.912	0.148

**Table 2 diagnostics-15-00354-t002:** Mann–Whitney U test.

	U	*p*	Rank-Biserial Correlation
Airway: Correct use of equipment	497.000	0.266	0.052
Airway: Recognition of structures	556.000	0.082	0.177
Airway: Diagnostic	566.500	0.092	0.199
Airway: Level of independence	585.500	0.028 *	0.239
Lung: Correct use of equipment	486.500	0.757	0.030
Lung: Structured examination	530.000	0.342	0.122
Lung: Recognition of structure	553.500	0.196	0.171
Lung: Diagnostic	607.500	0.048 *	0.286
Lung: Level of independence	596.500	0.06	0.262
Cardiac: Correct use of equipment	455.000	0.682	−0.037
Cardiac: Structured examination	512.500	0.567	0.085
Cardiac: Recognition of structure	602.500	0.07	0.275
Cardiac: Diagnostic	590.000	0.107	0.249
Cardiac: Level of independence	648.500	0.018 *	0.372
Duration of evaluation	336.000	0.082	−0.289

* *p* < 0.05.

## Data Availability

All data are contained within the article.

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
