# Peer review of "Transfer of POCUS Skills of Anesthesia Trainees from the Simulation Laboratory to Clinical Practice: A Follow-Up Pilot Evaluation After ABC US Protocol Training"

_diagnostics, 2025, doi:10.3390/diagnostics15030354_

Round 1
Reviewer 1 Report (Previous Reviewer 2)
Comments and Suggestions for Authors
The revised manuscript has met the requirement of publication at the journal of Diagnostics.
Reviewer 2 Report (Previous Reviewer 1)
Comments and Suggestions for Authors
The authors had addressed all my comments. I would like to recommend for publication.
This manuscript is a resubmission of an earlier submission. The following is a list of the peer review reports and author responses from that submission.
Round 1
Reviewer 1 Report
Comments and Suggestions for Authors
The manuscript titled "Transfer of POCUS Skills of Anesthesia Trainees from the Simulation Laboratory to Clinical Practice: A Follow-Up Pilot Evaluation After an A.B.C. US Protocol Training" explores the critical issue of Point-of-Care Ultrasound (POCUS) training and its transition from simulation-based education to clinical application. Given the growing reliance on POCUS in anesthesia and intensive care settings, this study is highly relevant. The findings highlight the impact of structured simulation training on the independence and diagnostic accuracy of trainees performing POCUS in real-world ICU settings. However, improved structuring of results, enhanced clarity in the evaluation methodology, and greater transparency in statistical analysis are necessary for the study’s conclusions to be fully validated.
Major Concerns and Recommendations:
1. Tables 1 and 2 contain valuable information but could be condensed for better readability.
2. A summary table that highlights only statistically significant findings would improve clarity.
3. Consider replacing tables with charts or graphical representations to enhance data visualization and accessibility.
4. The methodology lacks clarity regarding the assignment of scores (e.g., points 2 and 4) during trainee evaluations. A more detailed explanation of the grading criteria and examples of assessments would be beneficial to ensure reproducibility and transparency.
5. The calculation of the Mann-Whitney U test is unclear and does not allow independent verification of results. A step-by-step breakdown of the statistical analysis should be included, ensuring that reviewers and readers can replicate the findings. Given that the reviewer’s own calculations did not match the reported results, the hypothesis and conclusions of the study cannot be fully evaluated without further clarification.
Author Response
Robert Simon University of Oradea, Faculty of Medicine, Doctoral School, Oradea, Romania
Department of Anesthesia and Intensive Care, “Iuliu Hatieganu” University of Medicine and Pharmacy, Cluj-Napoca, Romania
Clinical Institute of Urology and Renal Transplant, Anesthesia and Intensive Care, Cluj-Napoca, Romania Str. Hameiului nr.6, Ap. 34, Cluj-Napoca, Romania
19.01.2025
Dear Reviewer,
Thank you for the review, comments and recommendations regarding the manuscript for the original research article entitled “Transfer of POCUS skills of anesthesia trainees from the simulation laboratory to clinical practice: a follow up pilot evaluation after an A.B.C. US protocol training “, submitted in “Diagnostics” on the 22nd of December 2024, with the manuscript ID: diagnostics-3415182. I would like to address here the comments and recommendations made by the reviewers and the changes that were made to the manuscript.
For the report made by Reviewer 1:
Comments 1: Tables 1 and 2 contain valuable information but could be condensed for better readability.
Response 1: Thank you for pointing this out, we agree that the visual aspect of the results was crowded and hard to read but the information in these tables is important. To address this issue Table 1 was transformed into a figure (Figure 2. in the manuscript) that contains graphical charts which show the scores attributed to each category evaluated for both simulation and no-simulation groups with the hopes that the information is more accessible and easier to read. Table 2 was named Table 1 and only the relevant information was kept making it more readable.
Comments 2: A summary table that highlights only statistically significant findings would improve clarity.
Response 2: To address this issue, we changed Table 1 into a figure with charts for better readability and condensed the information in Table 2 (which became Table 1) to make it mor easy and accessible. After having done the changes mentioned we decided to keep Table 3 (which was transformed in Table 2). Having done this we hope that the only the relevant information about the results are presented in the article and that it is more readable and accessible.
Comments 3: Consider replacing tables with charts or graphical representations to enhance data visualization and accessibility.
Response 3: We agree with this comment, and we changed Table 1 which contained the scores obtained for every scan performed in every category evaluated into a figure (Figure 2 in the revised manuscript) that contains charts highlighting the scores attributed for every scan in each category evaluated.
Comments 4: The methodology lacks clarity regarding the assignment of scores (e.g., points 2 and 4) during trainee evaluations. A more detailed explanation of the grading criteria and examples of assessments would be beneficial to ensure reproducibility and transparency.
Response 4: Thank you for pointing this out. We modifed the text accordingly and added supplemental information regarding the assignment of scores during trainee evaluations. These information can pe found in the Materials and Methods section of the revised manuscript in the “Protocol development” subsection.
Comments 5: The calculation of the Mann-Whitney U test is unclear and does not allow independent verification of results. A step-by-step breakdown of the statistical analysis should be included, ensuring that reviewers and readers can replicate the findings. Given that the reviewer’s own calculations did not match the reported results, the hypothesis and conclusions of the study cannot be fully evaluated without further clarification.
Response 5: To address this recommendation supplemental information was added and can be found in the section Materials and Methods, subsection “Statistical analysis”.
All changes done in the text of the revised manuscript are highlighted on a yellow background, including the tables.
Thank you for taking the time to review the manuscript and for your comments and recommendations.
Please address all correspondence concerning this manuscript to me at robert.simon@umfcluj.ro
Sincerely,
Dr. Robert Simon, MD, DESAIC
Anesthesia and Intensive Care consultant at Clinical Institute of Urology and Renal Transplant, Cluj-Napoca, Romania
Teaching assistant in the Department of Anesthesia and Intensive Care from “Iuliu Hatieganu” University of Medicine and Pharmacy, Cluj-Napoca, Romania
University of Oradea, Faculty of Medicine, Doctoral school – PhD. student
E-mail: robert.simon@umfcluj.ro
Reviewer 2 Report
Comments and Suggestions for Authors
This study aimed to evaluate the transfer of POCUS skills of anesthesia trainees from the simulation laboratory to clinical practice. The authors developed an evaluation protocol based on an ABC - US protocol and assessed trainees’ performance in the ICU. A total of 100 scans were evaluated, with 78 scans included in the final analysis. The results showed that trainees who attended prior simulation sessions generally had higher scores in most categories, with statistically significant differences in the level of independence regarding airway and cardiac evaluation and in the lung diagnosis category. The study concluded that initial POCUS simulation - based training using the ABC protocol is useful for transferring skills to the bedside and applicable in clinical practice.
The sample size, especially the number of trainees who had not attended the simulation sessions (only 15), is relatively small. This may limit the statistical power of the study and the generalizability of the findings. A larger sample would provide more robust results and a better understanding of the true differences between the simulation - trained and non - simulation - trained groups.
It is not clear whether the evaluators were blinded to the trainees’ simulation training status. Lack of blinding could introduce bias into the assessment, potentially affecting the objectivity of the results. Blinding of evaluators is an important methodological consideration to ensure the reliability of the study findings.
The evaluation was primarily centered on the trainees’ performance in using POCUS equipment, scanning techniques, and making diagnoses. However, the impact of these skills on actual patient outcomes, such as changes in patient management, length of stay, or mortality, was not assessed. Incorporating patient - related outcome measures would strengthen the clinical relevance of the study.
While the study evaluated the trainees’ ability to perform POCUS scans and make diagnoses, it did not explore how well the trainees integrated the POCUS findings into the overall management of the patient and the decision - making process. Understanding how POCUS is used in conjunction with other clinical information is crucial for its optimal utilization in patient care.
Although the study demonstrated some benefits of simulation training in terms of performance in a controlled clinical setting, it remains unclear how well these skills translate to the complex and dynamic environment of the actual ICU, where multiple factors and distractions are present. Further research is needed to assess the ecological validity of the simulation training.
The study did not account for individual differences among trainees, such as prior experience with ultrasound or different learning styles. These factors could potentially influence the effectiveness of the simulation training and the transfer of skills to clinical practice. Considering individual differences could provide more personalized insights and recommendations for training programs.
Figure 1 legend should be put below the plot.
Line 69 change “sings” to “signs”, thereafter
Author Response
Robert Simon University of Oradea, Faculty of Medicine, Doctoral School, Oradea, Romania
Department of Anesthesia and Intensive Care, “Iuliu Hatieganu” University of Medicine and Pharmacy, Cluj-Napoca, Romania
Clinical Institute of Urology and Renal Transplant, Anesthesia and Intensive Care, Cluj-Napoca, Romania Str. Hameiului nr.6, Ap. 34, Cluj-Napoca, Romania
19.01.2025
Dear Reviewer,
Thank you for the review, comments and recommendations regarding the manuscript for the original research article entitled “Transfer of POCUS skills of anesthesia trainees from the simulation laboratory to clinical practice: a follow up pilot evaluation after an A.B.C. US protocol training “, submitted in “Diagnostics” on the 22nd of December 2024, with the manuscript ID: diagnostics-3415182. I would like to address here the comments and recommendations made by the reviewers and the changes that were made to the manuscript.
For the report made by Reviewer 2:
Comments 1: The sample size, especially the number of trainees who had not attended the simulation sessions (only 15), is relatively small. This may limit the statistical power of the study and the generalizability of the findings. A larger sample would provide more robust results and a better understanding of the true differences between the simulation - trained and non - simulation - trained groups.
Response 1: Thank you for your comment. We agree that larger samples with similar numbers in the two groups would provide stronger statistical power to the study. Sample size differ between the two groups partially because the number of trainees that are currently working in our institutions that have not attended prior simulations sessions is small and availability to participate in the follow up study was low. We performed 100 scans and unfortunately only 78 were included in the final analysis because incomplete reports from the evaluators.
Comments 2: It is not clear whether the evaluators were blinded to the trainees’ simulation training status. Lack of blinding could introduce bias into the assessment, potentially affecting the objectivity of the results. Blinding of evaluators is an important methodological consideration to ensure the reliability of the study findings.
Response 2: Thank you for pointing this out and for the recommendation. The evaluators were blinded to the trainees’ prior simulation training status, and we added this in the revised version of the manuscript. This information can be found in the Material and Methods section, “Protocol development” subsection, lines 145-146, highlighted with a yellow background.
Comments 3: The evaluation was primarily centered on the trainees’ performance in using
POCUS equipment, scanning techniques, and making diagnoses. However, the impact of these
skills on actual patient outcomes, such as changes in patient management, length of stay, or
mortality, was not assessed. Incorporating patient - related outcome measures would strengthen
the clinical relevance of the study.
Response 3: We agree that we need to assess the impact of the simulation’s sessions and
trainees’ skills on actual patient outcomes, however our primary outcome was evaluation of
transfer of skills from the simulation laboratory to clinical practice and it did not involve any
intervention on the patient so patient outcomes were not evaluated.
Comments 4: While the study evaluated the trainees’ ability to perform POCUS scans and make
diagnoses, it did not explore how well the trainees integrated the POCUS findings into the overall
management of the patient and the decision - making process. Understanding how POCUS is used
in conjunction with other clinical information is crucial for its optimal utilization in patient care.
Response 4: We agree that integration of POCUS in clinical practice and patient treatment is of
utmost importance. Scans were performed on patients unknown to the trainees until the
moment they performed the scan. Future research is needed to clarify the utility of simulation,
transfer of simulation in clinical practice and patient outcomes.
Comments 5: Although the study demonstrated some benefits of simulation training in terms of
performance in a controlled clinical setting, it remains unclear how well these skills translate to
the complex and dynamic environment of the actual ICU, where multiple factors and distractions
are present. Further research is needed to assess the ecological validity of the simulation training.
Response 5: Thank you for pointing this out. We do understand that transfer of skills from the
simulation laboratory to clinical practice as well as transfer of skills from the training period to
actual unsupervised clinical practice is often done gradually and integrating these skills into daily
practice in a complex environment involving critical patients is sometimes difficult. We will try to
address this issue in future research regarding retention of skills and how well these skills
translate into clinical practice and patient outcomes.
Comments 6: The study did not account for individual differences among trainees, such as prior
experience with ultrasound or different learning styles. These factors could potentially influence
the effectiveness of the simulation training and the transfer of skills to clinical practice.
Considering individual differences could provide more personalized insights and
recommendations for training programs.
Response 6: Yes, we agree that individual differences among trainees regarding the style of
learning and skills acquisition can have an influence on their use of POCUS. Until this moment we
are not aware of a method to establish competency in POCUS in the critical care patient or
intensive care unit with heterogeneity among practice around Europe and the World. All trainees that undergo anesthesia and intensive care training in our institutions receive informal Point of Care Ultrasound teaching mostly by old ways of learning, like observation and supervised clinical practice. No curriculum regarding Point of Care Ultrasound, except the one presented by us in an earlier article is currently applied in our institutions thus we considered the experience of trainees to be somewhat homogenous. We demonstrated earlier in another published article cited in this manuscript the homogeneity of POCUS theoretical knowledge of trainees before attending the simulations sessions.
Comments 7: Figure 1 legend should be put below the plot.
Response 7: We did the change as requested and it is highlighted in the revised manuscript with a yellow background on lines 181-182.
Comments 8: Line 69 change “sings” to “signs”, thereafter.
Response 8: Changes were made to the typo in line 69 which was also present 3 more times.
All changes done in the text of the revised manuscript are highlighted on a yellow background, including the tables. Thank you for taking the time to review the manuscript and for your comments and recommendations.
Please address all correspondence concerning this manuscript to me at robert.simon@umfcluj.ro
Sincerely,
Dr. Robert Simon, MD, DESAIC
Anesthesia and Intensive Care consultant at Clinical Institute of Urology and Renal Transplant, Cluj-Napoca, Romania
Teaching assistant in the Department of Anesthesia and Intensive Care from “Iuliu Hatieganu” University of Medicine and Pharmacy, Cluj-Napoca, Romania
University of Oradea, Faculty of Medicine, Doctoral school – PhD. student
E-mail: robert.simon@umfcluj.ro